# Reducing Anxiety and Stress among Youth in a CBT-Based Equine-Assisted Adaptive Riding Program

**DOI:** 10.3390/ani12192491

**Published:** 2022-09-20

**Authors:** Kimberly Hoagwood, Aviva Vincent, Mary Acri, Meghan Morrissey, Lauren Seibel, Fei Guo, Chelsea Flores, Dana Seag, Robin Peth Pierce, Sarah Horwitz

**Affiliations:** 1Department of Child and Adolescent Psychiatry, New York University, New York, NY 10012, USA; 2Falk School, Syracuse University, Syracuse, NY 13244, USA; 3Mandel School of Applied Social Sciences, Case Western Reserve University, Cleveland, OH 44106, USA

**Keywords:** cognitive–behavioral therapy, equine-assisted services, adaptive/therapeutic riding, mental health

## Abstract

**Simple Summary:**

Reining in Anxiety (RiA) is a therapeutic program for youth with mild-to-moderate anxiety delivered in a therapeutic riding setting by Certified Therapeutic Riding Instructors. RiA is based on five foundational components of Cognitive Behavioral Therapy (CBT): in vivo exposure, cognitive restructuring, youth psychoeducation, relaxation, and caregiver psychoeducation about anxiety. The intervention sought to support youth between the ages of 6–17 with self-identified anxiety. Due to global pandemic trauma, in the second iteration of the protocol, researchers also included two evidence-based trauma components: maintenance and personal safety skills. All instructors were trained in the RiA curriculum and delivered the same lessons. In addition to assessing the youth’s perception and changes over time, the researchers also assessed changes in the horses, both through saliva sampling. The authors learned that RiA may be a promising approach for reducing anxiety and stress among youth and that the intervention can be delivered by adaptive/therapeutic horseback riding instructors in a non-clinic setting.

**Abstract:**

Reining in Anxiety (RiA) is a therapeutic program for youth with mild to moderate anxiety delivered in a therapeutic riding setting by Certified Therapeutic Riding Instructors. RiA was developed after a review of the evidence base for youth anxiety, is manualized, and includes five core CBT components: in vivo exposure, cognitive restructuring, youth psychoeducation, relaxation, and caregiver psychoeducation about anxiety. This study extended findings from a prior RCT that examined (1) the feasibility of collecting saliva samples from horses and children to measure stress (cortisol) and relaxation (oxytocin); (2) whether changes in stress and relaxation occurred both during each lesson and over the course of the 10-week intervention for horses and youth; (3) whether changes in anxiety symptoms, emotional regulation, and self-efficacy found in the first trial were comparable; and (4) if fidelity to the program was reliable. Youth participants (*n* = 39) ages 6–17 with caregiver-identified mild-to-moderate anxiety participated in a ten-week therapeutic intervention (RiA), which combined adaptive riding and components of CBT. Physiological data and self-report measures were taken at weeks one, four, seven, and ten for the youth and horses. Saliva assays assessed cortisol as a physiological marker of stress and anxiety, and oxytocin as a measure of relaxation. Fidelity data were recorded per session. Anxiety, as measured by caregiver self-reporting, significantly decreased from pre- to post-test, while emotional regulation scores increased. No significant changes in self-efficacy from pre- to post-test were observed. Saliva samples obtained from participants before and after riding sessions showed a consistent decrease in cortisol and a significant increase in oxytocin at two of the four timepoints (Week 1 and Week 7), but no overall pre- to post-test changes. Horse saliva data were collected using a modified bit; there were no significant changes in oxytocin or cortisol, suggesting that the horses did not have an increase in stress from the intervention. RiA may be a promising approach for reducing anxiety and stress among youth, as measured both by self-reported and by physiological measures. Collection of salivary assays for both youth and horses is feasible, and the intervention does not increase stress in the horses. Importantly, RiA can be delivered by adaptive/therapeutic horseback riding instructors in naturalistic (e.g., non-clinic-based) settings. As youth anxiety is a growing public health problem, novel interventions, such as RiA, that can be delivered naturalistically may have the potential to reach more youth and thus improve their quality of life. Further research is needed to examine the comparative value of RiA with other animal-assisted interventions and to assess its cost-effectiveness.

## 1. Introduction

Anxiety disorders are the most common class of mental health conditions in childhood and adolescence [1,2], affecting approximately 7.1%, or 4.4 million, youth between 3 and 17 years of age [3]. The prevalence of anxiety among youth has increased due to the SARS-CoV-2 pandemic [4]: However, even prior to the pandemic, youth screening positive for anxiety increased from 34.1% in 2012 to 44% in 2018 [5]. The median age of onset for anxiety disorders is 11 years, although this varies by the type of anxiety [6]. Females are more likely to have both an anxiety disorder and a lower median age of onset than males, which may be due to gender differences in processing serotonin [7].

Anxiety interferes with academic functioning and negatively impacts youths’ relationships with family members and peers [8]. Additionally, anxiety during childhood is associated with comorbid substance abuse and depression in adulthood [1,2]. The annual cost of anxiety disorders ranges from USD 43 to 47 billion dollars owing to mental health treatment, unneeded medical services, lower work productivity, and losses in earnings due to premature mortality [9].

Cognitive–Behavioral Therapy (CBT) is the evidence-based practice (EBP) considered to be the gold standard of treatment for youth with mild and moderate anxiety. In [10,11], the report that between 60% to 80% of children and adolescents show significant clinical improvement, and between 50% and 70% achieved remission following CBT. A meta-analysis including 19 studies found similar results; between 47.6% and 66.4% of youth achieved full recovery from anxiety when they received CBT compared to 20.6% and 21.3% assigned to an alternate therapy or waitlist control [12]. CBT has also shown long-term, post-treatment therapeutic gains of up to 16 years [13].

However, EBPs such as CBT are largely unavailable or inaccessible to most youth [14,15]. Families experience many barriers in accessing mental health services including transportation, competing priorities, and mistrust of mental health providers [16]. Workforce shortages affect the availability of services, and few mental health professionals are trained in EBPs [17]. State mental health authorities report that only 3% of their delivered services for youth are evidence-based [15]. Stigma persists as a significant barrier reducing the use of mental health services in traditional settings, such as clinics, health centers, and schools [18,19]. As a result, many families of children with mental health needs are unable to access needed services. Consequently, making effective services ecologically available to youth and families in naturalistic settings is a paramount public health goal. Animals have served as a supportive conduit towards building rapport and facilitating mental health treatment with youths. Through the increase in animal-assisted interaction research, practitioners now have resources to identify some mechanisms of change to facilitate therapeutic gains for youths [20,21]. Equine-assisted services (EAS) are a promising approach to reach populations that may be unable or unwilling to seek out therapies in traditional settings [22,23]. An increasing number of human services now incorporate animals, including equines, within non-traditional settings [22,23]. EAS is a term that refers to services provided by professionals who incorporate horses and other equines to benefit people. It can be provided by certified therapeutic riding instructors (CTRIs) or instructors without a certification. Therapeutic adaptive horseback riding often targets cognitive, physical, emotional, or social well-being [24]. Studies are beginning to show that EAS may be effective in reducing symptoms of post-traumatic stress among veterans and among other adults with a range of traumas [25]. However, few studies of EAS have investigated their effectiveness among youth with mental health conditions, nor have they used rigorous designs. Thus, understanding of the effectiveness of EAS for youth and the mechanisms by which it achieves any effects are very limited.

In 2018, the authors, who include Professional Association of Therapeutic Horsemanship International (PATH) Certified Therapeutic Riding Instructors (CTRI), equine owners, and child mental health research and treatment specialists, developed a novel adaptive riding program, drawing core elements from the comprehensive evidence-based reviews summarized by PracticeWise^®^ (Satellite Beach, FL, USA) on treatment for anxiety in youths. PracticeWise^®^ provides a virtual warehouse of synthesized research findings from more than 1000 controlled studies of effective psychosocial interventions for youth with mental health disorders, including anxiety. The adaptive riding program, called Reining in Anxiety (RiA), was designed specifically for youth with mild-to-moderate anxiety, with the intention of being delivered in an arena with fidelity by CTRIs, not licensed mental health clinicians [22,23,24,26,27].

RiA was initially tested in a randomized trial in New York City with a sample of youth between 6 and 16 years of age who met criteria for mild-to-moderate anxiety [28]. Youth who received RiA showed significant improvements in anxiety and self-efficacy over the comparison group, which received standard therapeutic riding [22]. This paper describes findings from a second study designed to expand upon the initial trial of RiA. This second study was conducted at a different equine facility, in a different state, with a new sample of youths and caregivers, and a different group of riding instructors who were trained to deliver the intervention. The sample from the current study was similar to the original in that the participants were drawn from the same age range (6–17 years old) and had a similar ethnic and racial breakdown. The current study had five goals: to assess (1) the feasibility of collecting saliva samples from horses and children to measure stress (cortisol) and relaxation (oxytocin); (2) whether salivary assays of cortisol as a physiological measure of stress/anxiety and oxytocin as a measure of relaxation/bonding changed both during each lesson and over the course of the ten weeks for horses and youth; (3) whether changes in anxiety symptoms, emotional regulation, and self-efficacy found in the first trial were comparable; and (4) whether fidelity to the program was reliable. Saliva assays to examine whether horses experienced increased stress due to their participation in the study were collected to respond to calls for increased attention to animal welfare in animal-assisted therapies [29,30]. Assessing fidelity was a priority given that non-mental health professionals conducted the sessions.

## 2. Methods

### 2.1. Participants

Participants were 39 youth between 6 and 17 years of age with mild-to-moderate anxiety, as measured via self-reports from their caregivers. All participants were active students at Fieldstone Farm Therapeutic Riding Center (FSF), a premiere Professional Association of Therapeutic Horsemanship International (PATH)-accredited facility in northeast Ohio. The Center is located on 45 acres, with 40 horses and 19 CTRIs. A CTRI is a credentialed professional through PATH International who provides adaptive/therapeutic horseback riding lessons, mounted or unmounted, to individuals with special needs. Each lesson had 2–4 students within a reasonable developmental age, and the instructor had the skill to successfully teach the group. Lessons were pre-established based on initial enrollment at the Center and were maintained throughout the study. Adaptive horseback riding lessons are often supported by 1–3 volunteers per rider.

Screening for study eligibility was conducted by FSF using the Generalized Anxiety Disorder 2-item instrument (GAD-2) [31] and the Children’s Global Assessment Scale (CGAS) [32]. Briefly, the GAD-2 consists of two core anxiety symptoms, and the respondent endorses how often the child has experienced those symptoms in the last two weeks on a 4-point scale (from 0 = not at all to 3 = nearly every day). Scores range from 0–6; a cutoff score of 3 indicates a possible case of generalized anxiety disorder and that further evaluation is warranted [31]. The CGAS is a 1-item measure of functional impairment and the overall severity of disturbances in multiple domains, with scores ranging from 1 to 100, with higher scores representing better functioning [32].

Inclusion criteria were English-speaking youth between 6 and 17 years of age who had a score of 2 or greater on the GAD-2 and a score of 41 or higher on the CGAS based upon caregiver reports. Exclusion criteria included youth who did not meet the inclusion criteria (e.g., were under the age of 6 or over the age of 17), had a score below 2 on the GAD-2, or did not meet a minimum level of functioning (did not meet a score of 41 or higher on the CGAS). Inclusion criteria for caregivers included being aged 18 years or older and speaking English; exclusion criteria for caregivers included being younger than 18 years of age or unable to provide informed consent. Staff from FSF were trained by a member of the research team to screen potential participants and refer caregivers of eligible participants to the NYU study team for more information about the study.

### 2.2. Recruitment

Recruitment and delivery of the intervention occurred between January and May of 2021. Of the 43 youth/caregiver dyads approached, 39, or 91%, agreed to participate. Data were collected in weeks one (pre-test), four, seven, and ten (post-test). Since volunteers are an integral part of adaptive riding lessons, all volunteers participating in RiA were provided information about the study.

Of the 40 horses homed at the Center, 8 horses were engaged in the study. As a premier PATH-Intl.-accredited center, all horses are under the care of a trained equine director. The Center meets the ten equine welfare and management standards, in addition to equine skills, facilities, and other areas of assessment [33]. All horses are assessed individually for their needs and are cared for to ensure ethical wellbeing and quality of life. Additionally, the Center is endorsed by Pet Partners, the nationally recognized gold standard for Animal-Assisted Interventions and Professionals.

### 2.3. Procedures

The New York University (NYU) Langone Health Institutional Review Board (IRB) and the Biomedical Research Alliance of New York LLC (BRANY#183361) IRB approved the protocol. Potential participants were recruited through conversation with existing FSF program attendees and advertisements posted on the FSF webpage. Caregivers and potentially eligible youth met with the study’s research coordinator, who explained the study in detail and secured written informed consent from the caregiver and assent from the child. After consent and assent was obtained, eligible participants were assigned to RiA, an adaptive riding group intervention, with up to four youth per group.

### 2.4. Intervention Development

The authors of this paper, all of whom are either CTRI professionals, mental health services researchers, and/or licensed mental health professionals, developed a 75-page manualized program for Reining in Anxiety (RiA). RiA is not intended to be a treatment for a diagnosis or disorder; rather, the intervention aims to teach skills that may mitigate the symptoms of anxiety. The intention is to teach skills that youth find helpful and that can translate to other environments. It includes both mounted and unmounted equine interaction activities to develop horsemanship skills, but also adds an intentional focus on mental health goals. Development of the content of the intervention was based on information drawn from PracticeWise^®^ [34], a website that contains synthesized research findings from randomized controlled studies of effective psychosocial therapies for children and adolescents with mental health conditions. In a search of the PracticeWise^®^ database, the focus was on treatments for anxiety disorders. The top-five most effective components, or those with the “best supported research” were: in vivo exposure (92%), cognitive restructuring (66%), client psychoeducation (56%), relaxation (44%), and caregiver psychoeducation (40%). PractiseWise^®^ provides “practitioner guides” for each of these core elements, which were utilized by authors to develop session content [34]. Additionally, the current study included components for “best supported research” in treating trauma in youths. There was significant overlap in the top effective components, but authors included maintenance/relapse prevention and personal safety skills, which were both found to be effective treatments for trauma in youth in 50% of protocols reviewed in the PracticeWise^®^ database. Authors specifically excluded the top effective core element of narrative creation due to the group setting and use of non-mental-health-providers in its delivery. Authors also consulted with PraciceWise^®^ providers on the structure and sequencing of core elements.

The identified components were integrated into ten, 45 min adaptive/therapeutic riding sessions, detailed in a comprehensive intervention manual, with delivery supported by a separate set of implementation props. Each 45 min session focused on skill development of a specific component of CBT (e.g., restructuring negative thoughts) using repetition, and was reinforced through a weekly homework journal to promote the generalization of skills outside of sessions. While the instructors were responsible for delivering the content of the RiA session, they also had time to teach horsemanship skills in alignment with their role as CTRI, adaptable to riders’ current skill level. The caregiver psychoeducation component was delivered in 7 min segments at the end of each session, with the intention to relay information to the caregiver, as well as an opportunity for instructors to assess children’s skill and knowledge retention. All learning was supported by both a website [32] and written psychoeducation for caregivers. A detailed description of the development of this intervention is described in more detail in Acri et al. [27].

### 2.5. Training

Eight FSF CTRIs were trained in delivery of the RiA protocol. The PATH CTRIs had over ten years of teaching experience with FSF and attended a three-day, in-person training provided by a co-author and co-developer, who is both a PATH CTRI and a licensed mental health clinician. Training the CTRIs in the intervention consisted of an overview of the therapeutic stance, practice of CBT skills (such as teaching relaxation, exposure, and providing feedback), small group discussion, and content evaluation to ensure material retention. Instructors also received extensive implementation supports, including props, such as cards outlining each session’s components and fidelity checklist cards for instructors to assure that essential elements of that session were completed.

Instructors participated in weekly peer study groups starting the week following the training, which continued for the duration of the study. The instructors also met weekly for supervision with the co-author and co-developer of RiA during the active portion of RiA. Instructors participated in providing feedback on RiA via survey and focus groups after completion of the protocol trial.

The Research Coordinator, located on-site at FSF, is a Licensed Social Worker and CTRI with five years of teaching experience, four of which were at FSF. She was also trained as an Equine Specialist in Mental Health and Learning through PATH Intl. The Research Coordinator served as the Program Director for the Center, and as such, also provided weekly supervision to the instructor, but did not teach any weekly intervention sessions.

### 2.6. Fidelity

Fidelity to the manualized protocol was assessed throughout the intervention. The fidelity checklists from the initial study were used for each of the ten sessions, with an adaptation to include the addition of the trauma components. The checklist identified key elements of the session the instructors needed to complete. An example checklist item is: “Instructor gave riders three chances each to provide examples of the connection between a thought, feeling, and action.” The number of items on the checklist for each session ranged from 13 to 18 items.

Checklists were completed by direct observation. A research assistant checked a binary yes/no rating on whether or not the instructor completed that part of the session, and fidelity was scored by calculating the percentage of elements marked ‘yes’ in each session. Research assistants were trained to observe the lessons from outside the arena, where they could see and hear the lesson, but would not be a distraction to the riders or horses. Of the 141 lessons/sessions completed, 136 were rated for fidelity (96.5%). Fidelity scores per individual session (i.e., per week) ranged from 88.4% to 100%. Average fidelity scores by instructor ranged from 92.5% to 100%. The average fidelity percentage across all sessions was 97.14%.

## 3. Measures

Youth were assessed via caregiver reports pre-test and at the end of the 10-week session on symptoms of anxiety (i.e., using Screen for Child Anxiety Related Disorders, or SCARED), attachment anxiety (i.e., using a modified Brief Experiences in Close Relationships, or ECR, to assess children’s close relationships), and self-efficacy (i.e., using a modified Self- Efficacy Questionnaire for Children, or SEQ-C, to assess emotional self-efficacy). Caregivers also provided sociodemographic data at baseline. Baseline measures were administered in person using iPads, with caregivers answering through REDCap (Research Electronic Data Capture, or REDCap, is a secure web-based application used to build and manage online surveys and databases). Post-survey measures were sent to caregivers via email and completed online at home via REDCap. For details on the psychometrics of the psychological measures, see Acri et al. [27].

## 4. Physiological Data

Saliva samples were collected in weeks one, four, seven, and ten from youth participants before and after each session. Saliva samples were also collected from the horses if the student rode one of the eight horses identified for engagement in the study. Youth used a sterile vial and straw to collect pooled saliva. Due to the COVID-19 pandemic, caregivers were encouraged to collect their child’s saliva while wearing sterile gloves. The Research coordinator and research assistants provided verbal directives. If students were unable to generate 1 mL of saliva or were unable to collect pooled saliva, researchers allowed the use of a cotton swab to be placed under the youth’s tongue in order to collect a specimen.

Using a similar process, the horses’ saliva samples were collected before and after their lesson at weeks one, four, seven, and ten. Each horse had a bitless bridle, with the modified bit attached where a typical bit would be. The modified bit was created based on Contreras-Aguilar’s [35] design: the tubing, made of pliable rubber, had slits cut to allow for the collection of pooled saliva [36]. Health data regarding the horses’ health, age, and typical workload were also tracked to holistically understand the physiological data.

Using sterile gloves, clean cotton gauze was inserted prior to collection. The horse wore the bit for 90 s before the lesson and for 90 s after the lesson. The Research Coordinator was solely responsible for the collection of the saliva samples from the horses. All saliva samples for both children and horses were marked with unique identifiers. Once saliva samples were provided by both humans and equines, they were stored in a privately secured and locked freezer.

## 5. Data Analyses

Univariate statistics were used to describe the sociodemographic characteristics of all participants including children and caregivers. Paired t-tests were used to assess pre/post differences in anxiety (SCARED), emotional regulation (ECR), and self-efficacy. A linear regression was conducted to test if the pre/post changes differed by age. A two-sample t-test was conducted to test if pre/post changes differed by gender, and an ANOVA was conducted to test if pre/post changes differed by race. Linear mixed-effects models were performed to examine changes in oxytocin and cortisol of the participants over time at week one (pre-test), four, seven, and ten (post-test). Wilcoxon signed-rank tests were performed to analyze the pre/post changes in oxytocin and cortisol of the kids at each of the four timepoints. The same analyses were performed for horses. Mean values were used for the horses with different riders at each time point. All analyses were conducted with the software R.

## 6. Results

### 6.1. Sample Characteristics

The sample consisted of 39 youth/caregiver dyads. Children were 11.5 years of age on average (*SD* = 2.8), and slightly over half were female (*n* = 20, 51.3%). The majority of children were white (*n* = 31, 79%), followed by 5 (12.8%) who identified as Black/African American. All identified as non-Hispanic. Almost two-thirds had a mental health diagnosis (*n* = 24, 62%), which is representative of the individual students who engage in weekly lessons at the center. Caregivers were 45.3 years of age on average (*SD = 6.33*), primarily female (*n* = 31, 317, 79.5%), White (*n* = 37, 94.9%), and married (*n* = 30, 81.1%). Over one-third (*n* = 16, 41%) completed a graduate or professional degree, and over half (*n* = 20, 51.3%) were employed full-time. The majority of families had an annual family income of over $50,000 USD (*n* = 29); ten families stated income of less than $50,000 USD.

### 6.2. Psychological Outcomes for Youth

There was a significant difference in anxiety levels as measured by the total SCARED score from pre-test to post-test. Pre-test, the scores ranged from 42 to 101 (M = 66.28, *SD =* 16.46), indicating the likely presence of an anxiety disorder. Post-test, the scores ranged from 40 to 97 (M = 59.65 SD = 12.18), a statistically significant reduction in symptoms of anxiety (*p* = 0.001). In terms of emotional self-efficacy, pre-test, this subscale score ranged from 8 to 39 (M = 20.87, *SD* = 7.13), and post-test, it ranged from 8 to 40 (M = 23.14, *SD* = 7.40), indicating moderately low levels of self-efficacy. There was no statistically significant difference between self-efficacy scores from pre- to post-test (*p* = 0.07), although the trend was in the expected direction.

Emotional self-regulation as assessed with ERC pre-test ranged from 32 to 77 (M = 55.3, *SD* = 12.62) and from 32 to 78 post-test (M = 58.16, SD = 12.58); this change was statistically significant (*p* = 0.033), indicating an improvement in emotional regulation post-intervention. A linear regression was conducted to test if the pre/post changes differed by age, and there was no statistically significant difference. A two-sample *t*-test was conducted to test if pre/post changes differed by gender, and again, there was no statistically significant difference. An ANOVA was conducted to test if pre/post changes differed by race. While the SCARED and SEQ-C were not significant, race did predict pre/post changes for the ERC (*p* = 0.034).

### 6.3. Physiological Outcomes for Youth

Saliva samples obtained from participants before and after the riding intervention at the four time points (weeks one, four, seven, and ten) showed a consistent decrease in cortisol, along with a significant increase in oxytocin at two of the four timepoints (Week 1 and Week 7). In weeks one and four, cortisol and oxytocin levels remained the same, while week seven showed a decrease in cortisol and an increase in oxytocin, and week ten showed a decrease in cortisol and an increase in oxytocin (see Table 1 and Figure 1). However, overall pre/post differences across the study did not yield significant changes for either cortisol or oxytocin.

### 6.4. Physiological Outcomes for Horses

To mirror the analysis of the riders, linear mixed-effects models were performed to examine changes in oxytocin and cortisol in the horses over the four timepoints and over the entire 10-week period from pre- to post-test. There were no changes pre- to post-test during any of the weeks. A Wilcoxon signed-rank test showed no significant changes in oxytocin or cortisol. Over the entire course of the 10-week period, while cortisol increased, there was no statistically significant change (*p* = 0.203). Similarly, oxytocin was consistent, and there were no statistically significant changes (*p* = 0.936).

## 7. Discussion

This study examined the effectiveness of a CBT-based equine-assisted therapeutic riding program targeting youth with mild-to-moderate anxiety. The initial trial of Reining in Anxiety (RiA) used a randomized controlled design and found that RiA reduced symptoms of anxiety and increased perceptions of competence or self-efficacy in managing symptoms more than standard adaptive/therapeutic horseback riding instruction [23,27].

The study reported in this paper was conducted at a different equine facility in a different state focused on recreating findings related to changes in psychological outcomes (i.e., anxiety, emotional regulation, and self-efficacy) and on examining with exploratory analyses the feasibility of collecting biomarkers for anxiety, stress, and relaxation through saliva assays.

Saliva assays assessed cortisol as a physiological marker of stress and anxiety and oxytocin as a measure of relaxation. Specifically, the interest was in exploring the feasibility of collecting saliva samples from horses and youth in order to observe whether changes occurred and, if so, whether these changes followed a similar pattern to the psychological changes in self-reported anxiety over the course of the 10-week program. We also aimed to assess the patterns of these physiological changes during each lesson. Because there is increasing attention to the bi-directionality of protections for human subjects as well as animal participants in research, particularly in studies of animal-assisted interactions, a secondary goal of the research was to examine whether horses experienced an increase in stress due to their participation in the study [29,30]. This is in ethical obligation in alignment with the call for equine welfare assurances in research and practice [37,38,39].

Since self-reported assessments of youth anxiety can have inherent biases, we included objective measures of stress, anxiety, and its opposite (i.e., relaxation) to see if the same trends would be observable. Pooled saliva (e.g., spit) was used to collect salivary assays from youth, and a modified bit [36] was used to collect the same from the horses.

Saliva was collected at four timepoints throughout the ten weeks of the program and, therefore, could be used to assess changes within each of those four lessons, as well as across the entire program. Among the youth sampled, we found a consistent decrease in cortisol and increase in oxytocin at two of the four timepoints (week 1 and week 7), and no change in the other two.

Overall trends from pre-test to post-test for youth showed no changes. The changes in two of the four individual timepoints may suggest that the program was influencing physiological levels of stress during some lessons and not others. Because of the small sample size, it is also possible that the study was underpowered in detecting an effect. The lack of significant changes from pre-test to post-test in these physiological measures was disappointing, but not surprising. RiA was 45 min per week, a fraction of time in the context of a youth’s daily life, so it would likely require a larger intervention over a longer period of time to influence levels of cortisol. Further, it is well known that cortisol levels change frequently over the course of a single day, so consistent and measurable changes over a ten-week period would be challenging to find.

A welcome finding was that among the horses, we found no change in stress or relaxation. This held true both for the individual four timepoints as well as across the ten-week program. This suggests that the riding program itself did not add stress to the horses’ daily routine.

Fidelity to the program’s core components when delivered with supervision and supports by non-mental health providers, specifically PATH CTRIs, was assessed and found to be excellent: instructors achieved an average score of 97.14% across sessions. This was almost identical to the high-fidelity ratings for this intervention in our first study [1]. Reproducing these fidelity ratings was noteworthy, as this group of eight instructors was delivering this intervention for the first time. In the original study, two instructors delivered the intervention several times over multiple 10-week sessions, and so had much more experience and practice. These fidelity findings suggest that training non-mental-health-professionals to provide targeted interventions with supervision and support can be beneficial to youth mental wellness. Taking established therapeutic modalities from the evidence base and adapting them for delivery in a variety of alternative environments (such as riding stables) in order to alleviate distress among adolescents is an important public health goal.

As with our prior study, caregiver reports of youth anxiety significantly decreased from pre-test to post-test, and emotional regulation significantly increased. This suggests that the 10-week program seemed to benefit youth. There were no significant changes in self-efficacy, unlike in the prior study, although the trend was in the expected direction. This may be due to factors that could not be controlled, such as the level of prior experience managing anxiety, prior experience riding, length or intensity of the program, or measurement error due to the appropriateness of the scale for this sample (i.e., the scale was not specific to CBT self-efficacy).

### Limitations

The sample for this study was largely white (79%), and thus, the findings cannot be assumed to generalize to non-white samples. A lack of access to adaptive/therapeutic riding programs for youth of racially diverse backgrounds is a limitation. It is encouraging to note that Fieldstone and a growing number of EAS programs now offer scholarships to support programming for families who may be facing financial hardships, and this often includes families from diverse racial/ethnic backgrounds.

Because this was a small sample (*n* = 39) and involved exploratory analyses, it was underpowered to detect some effects and, unlike our original study of RiA, did not include a comparison group. While the optimal timepoints for saliva collection are not established, it is feasible to collect the same assays from both horses and humans. Moreover, we found that doing so does not add stress to the horses. Similarly, the participants’ typical riding time was not augmented to adjust for ideal saliva collection time points, but rather, they continued riding at their scheduled time (between 4 p.m.–8 p.m. ET), when cortisol’s diurnal rhythm varies [40,41,42]. While cortisol was included, we also acknowledged that the analyte is not as sensitive a marker of stress as other assays, such as alpha-amylase. Cortisol sampling in isolation of other analytes and other forms of data is not a sufficient measure of definitive changes in stress. However, alpha-amylase was unreadable in our equine samples. Thus, the alpha-amylase data were disregarded from the planned triad analysis of physiological data.

Additional limitations include that only intervention fidelity was measured in this study. Additional research is needed to confirm that not only was the skill delivered, but that the instructors taught the skill correctly. While there was live observation during sessions, the sessions themselves were not video/audio-recorded, and thus not independently coded for interrater reliability.

## 8. Conclusions

Results from this study, combined with our prior RCT study, suggest that RiA may be a promising approach for youths aged 6 to 17 in reducing symptoms of mild-to-moderate anxiety and improving emotion regulation. High fidelity to the protocol is possible in therapeutic riding environments. The fact that RiA can be delivered by adaptive/therapeutic horseback riding instructors with fidelity is important, because most children and adolescents with anxiety rarely receive treatment. Traditional mental health services are difficult to find, have long waiting lists, and are typically not evidence-based. EAS that have data to corroborate their effectiveness and that can be delivered by CTRIs can reach a much larger group of children and adolescents who suffer unnecessarily from anxiety. There are over 873 therapeutic riding centers accredited by PATH Intl. employing more than 4800 certified instructors and equine specialists [33]. Given the prevalence of youth suffering from anxiety, broadening the reach of effective mental health services into non-specialty community services can have a two-fold benefit: it can expand the very limited mental health workforce currently available, and it can increase access to effective services for more youth.

Further research on this and other alternative evidence-based services are needed to broaden the reach of effective programs into non-traditional mental health settings, including stables. Cost-effectiveness studies are needed to determine whether the benefits of these programs outweigh the costs to families and to stables. Such studies could also assess whether participation in these kinds of programs reduces the need for or intensity/dosage of medication treatments. Comparative studies of animal-assisted interventions are also needed to better understand the mechanisms of action, the relative effectiveness of them vs. other types of therapies, and the associated costs.

The COVID-19 pandemic, coupled with an increased prevalence of youth mental health challenges, including rising rates of suicide, depression, and anxiety, speak to the looming public health crisis for the next generation. A recent Surgeon General’s Advisory on Youth Mental Health [43] called for the expansion of effective services into routine and non-traditional community settings, such as equine centers across the USA. Targeted attention to expanding research on the non-traditional delivery of effective mental health services to children and youth is especially critical right now. Developing a broader array of effective community services to offset the alarming rise in youth mental health problems should become a national priority.

## Figures and Tables

**Figure 1 animals-12-02491-f001:**
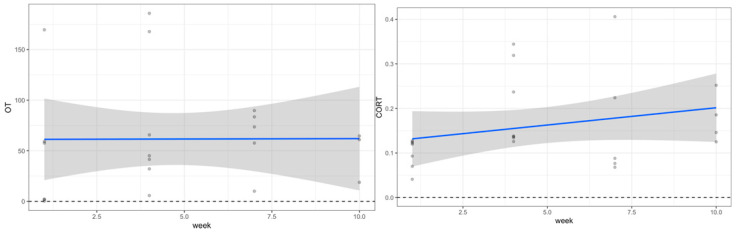
Oxytocin and cortisol change from week 1 to 10 with trend line.

**Table 1 animals-12-02491-t001:** Pre/post changes for oxytocin (OT) and cortisol (CORT) of youth and horses.

	Analyte	Youth		Horses	
		Median of Differences (Pre–Post)	*p*-Value	Median of Differences (Pre–Post)	*p*-Value
Week 1	OT	13.03	0.016 *	−9.09	1.000
	CORT	−0.04	0.009 *	−0.01	0.297
Week 4	OT	7.21	0.202	30.6	0.297
	CORT	−0.04	0.008 *	0.06	0.297
Week 7	OT	3.93	0.050 *	51.98	0.125
	CORT	−0.04	<0.001 *	−0.01	0.813
Week 10	OT	3.66	0.328	32.37	0.250
	CORT	−0.04	<0.001 *	0.03	0.625

* Refers to *p*-value < 0.05, those that are statistically significant.

## Data Availability

All data can be accessed by emailing the corresponding author.

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
