# Peer review of "Reducing Anxiety and Stress among Youth in a CBT-Based Equine-Assisted Adaptive Riding Program"

_animals, 2022, doi:10.3390/ani12192491_

Round 1
Reviewer 1 Report
Thank you for your work on CBT Based Equine Assisted Adaptive Riding (RiA program). I enjoyed reviewing this article and found it to be a great addition to the literature on these important interventions.
I was very excited to read that you also made sure to asses samples of the horse saliva. It is relieving to see that the study of these equine assisted interventions are measuring and consideration of the stress to the horses.
Reading through the study procedures, I appreciated the detail of the explanations of how samples were collected for both the children and the horses.
I noted the following areas for possible minor revision:
Line 128 and Line 132 in the citation for Hoagwood et al., the year is stated as: "20217" I believe that may have meant to be 2017 if I am understanding this correctly.
On page 16, line 613, I believe there is an extra "(" that needs to be removed between the comma after 23.14 and SD.
Lastly, at the end of the article, around page 21 lines 789 to 799, it may be beneficial to address the limitations of access to such a therapeutic riding program for children from families with lower income and less access to transportation to such riding centers as well as caregiver access (for example, many children relying on a sole single parent, may be less likely to be able to get the access to a program like this, due to the single parent being stretched too thin to take to and afford financially such a program). It may also be worth mentioning then how this impacts access to children who are not white.
Overall, I'm very excited about this work and this article. Thank you again for your work and contributions to the field!
Author Response
Please find our letter to reviewers attached

Reviewer 2 Report
Dear Authors, thank you for the opportunity to review your manuscript and provide feedback. The topics of reproducibility, fidelity, feasibility, and equine welfare for Equine-Assisted Services are much needed, and I commend the researchers for doing work in these areas. My primary concerns with the manuscript are outlined below. I hope the authors will find this information helpful for future revisions. In order for this to be a helpful and compelling article on replication and extension, the authors should revise the manuscript to make clear comparisons between data sets, introduce new information that constitutes meaningful extension of the work, and provide a better synthesis in the discussion with specific implications for research and practice. Currently lacking in this manuscript is a clear description of how the initial study was extended, and so it is not clear that this constitutes a true extension of the work. Aspects of replication should be provided in more detail as well. For guidance, I recommend
Brandt, M. J., IJzerman, H., Dijksterhuis, A., Farach, F. J., Geller, J., Giner-Sorolla, R., ... & Van't Veer, A. (2014). The replication recipe: What makes for a convincing replication?. Journal of Experimental Social Psychology, 50, 217-224.
There appears to be a disconnect between the conceptual basis and rationale for the intervention and the type of service delivery approach/ intended purposes (i.e., adaptive riding). Could the authors provide more justification on this? In other words, can it be called a therapeutic riding service when mental health interventions and goals are included as the focus rather than providing people with disabilities access to horses and learning riding skills? With this in mind, the article lacks evidence to support the rationale for the intervention approach, which impacts the authors’ argument for a need to test the intervention’s replicability. My question to the authors: How might the authors address the ethical and scope of practice issues in training therapeutic riding instructors to incorporate CBT into adaptive riding/mounted activities?
· The article introduction/literature review could benefit from revisions to the logic chain of the argument, and more literature to link the central argument for the concept, research design, instrumentation, and main objectives to replicate and extend. I also recommend the authors reduce the number of self-citations in this section of the manuscript, as a more in-depth literature dive would help strengthen the rationale for this study.
· I recommend the authors narrow the focus of the article. Measuring equine stress was not fully fleshed out in this manuscript in the introduction, methodology, analyses, results, or implications.
· I was not able to find information regarding ethical approval from an institutional animal care and use committee to collect cortisol from horses with a novel device.
· In terms of recruitment, please describe the rationale for the age range chosen for this study as the range represents several developmental stages with different needs related to teaching CBT-based skills. It was not clear if six-year-old children participated in sessions with children in different developmental stages, nor was it clear what the group size was for a riding session. How was recruitment the same or different from the initial RiA study. In terms of extending the original work, how might this be achieved in terms of sample size composition and subsequent analyses?
· In regards to reproducibility, was session composition the same as the initial study, or were adaptations made? Regarding the fidelity checklist, was this a new checklist, or the same checklist already developed for the initial study? The way it is worded now suggests a new checklist was developed, which would be important to note, and changes what can be compared with the initial study.
· The article was missing key elements for reporting fidelity and feasibility. For example, it is unclear if ratings of interrater reliability were collected from people scoring the sessions for intervention fidelity. Also, given that cortisol samples have been successfully collected from children in previous equine-assisted services studies (see Pendry et al.), what aspects of feasibility were the researchers intending to evaluate in regards to cortisol measurement with the child participants? More generally speaking, it would be helpful if the authors could flesh out the specific areas of feasibility they aimed to address, the specific measures related to feasibility, and how their results did or did not address those areas.
·
· The authors noted the horses were not stressed; however, this a problematic conclusion given they did not include behavioral observations in the analyses.
· Review for typos (e.g., typos in in-text citations) and formatting issues throughout the manuscript. I am not sure if some of these issues were related to transferring a word document to PDF.
Author Response

(The authors gave the same response as above.)

Reviewer 3 Report
Hi many thanks for your paper I enjoyed reading it I have some suggested changes.
* You need to include a section on how the horses are kept and cared for within your ethics section to ensure that their wellbeing in terms of best practice for keeping horses is complied with.
*You do not reference `Practice Wise` properly in the reference list, or the paper which gives more detail about the development of the intervention Acri et al., I feel that you need to be very transparent about the characteristics of the intervention and should include references for all the research used to inform this development. In addition you mention the `top 5 most effective components` again you need to offer full information on what these are and what resources were used to inform them.
* You also need to include the `specific components of CBT used`
* Your findings need to be clearly summarised together in the paper to help the reader clarify what you have discovered as you have multiple data sets.
*Your paper would benefit from a limitations section.
*On line 525 you state `individual bitless bridle with modified bit attached` this does not make sense you need a clearer description or diagram to help the reader fully understand what you did.
*Table 2 you mention (post - pre) do you mean (pre - post)?
* You have varied gaps between lines in your document which need to be consistent throughout.
Author Response

(The authors gave the same response as above.)

Round 2
Reviewer 2 Report
Dear Authors,
Thank you again for the opportunity to review your manuscript, and I appreciate the groups’ thoughtful and detailed responses to the initial round of comments which helped me to better understand the intent and direction the authors’ wished to take with this research. Please see my additional comments and questions below.
Please review for minor spelling and typo errors.
Simple summary says children 6-17, the rest of the manuscript says 6-16. Please reconcile.
Line 32: Recommend a word such as “comparable” or “similar” instead of re-created.
Please describe how many children were in each adaptive riding lesson, including information on if they were grouped by similar ages. Was a 10-year-old in a group with a 16-year-old? Or, were these private lessons?
Beginning on line 98: The authors have not fully defined the term therapeutic/adaptive riding. The purpose for which is to adapt the environment, approach, and/or equipment to facilitate riding for those who would otherwise experience barriers. The article does not mention how the CBT curriculum facilitated participant ability to learn riding skills, but rather focuses primarily on general symptom reduction as an outcome. It is critical that the authors clarify that adaptive riding lesson are not for the purposes of treating a disorder. I share this feedback in particular because there was no measurement related to whether they increased their riding skill attainment as an outcome of their anxiety being reduced. While this reviewer can appreciate the role of paraprofessionals for increasing access to mental health treatments, and CTRIs might fit this description someday, the authors are describing what could appear to be a different service from adaptative riding. This complicates the public’s understanding of the services. At minimum, this should be clarified somehow in the article. This reviewer also understands the challenges our industry experiences with terminology and feels it is critical to provide boundaries around what a service is intended to do. If CTRIs were trained to use CBT to enhance children’s ability to ride through anxiety, this should be clarified to support the program being labeled as therapeutic/adaptive riding/horsemanship.
From PATH, Intl: “Services that are specifically focused on adapting groundwork and riding experiences to be accessible to individuals and groups with diverse needs. Provided by specially trained and certified equine professionals, therapeutic/adaptive horsemanship helps participants attain individualized horsemanship skills and experience many naturally healthful benefits of horseback riding and other horsemanship activities”.
Table 1 layout: Recommend not using x or y format. Some descriptive could be written in text so the table remains uniform. Some already are, and therefore it is not necessary to include in the table as well. The lines and heading are confusing. For instance, race looks separate from the data it is supposed to be grouped with.
Please indicate if all children in the sample completed all 10 of the sessions. Were there any gaps in attendance? Did the children receive a make-up if this occurred, or did some children complete less than 10 sessions?
In the results section, the authors state that horse cortisol numbers increased, but not significantly. Lack of significance does not necessarily mean something important did not occur. Could the authors please contextualize the readings for horses, as well as children. What do the numbers mean? What reading would indicate lack of stress and what reading would indicate presence of stress?
In the limitations section, starting line 465, please include citations regarding information on what can be expected from cortisol sampling.
A limitation should be noted that cortisol sampling in isolation is not sufficient to measure changes in stress. Behavioral analysis should also be included. Therefore, it is unclear if the horses were stressed already. This relates to my previous comment asking for the authors to contextualize what the cortisol readings meant. I appreciate the authors clarifying that the results do not mean the horses were not stressed; however, It is this reviewer’s opinion that saying the program did not increase stress might also lead to more questions from the reader, such as was the horse stressed, but did not get worse?
Another limitation that should be included is that only one type of fidelity was measured (intervention delivery), therefore, more research is need to confirm that not only was the skill delivered, but that the instructors taught the skill correctly.
Consider including in your limitations that live observation was used and sessions were not independently coded for interrater reliability.
The authors need a clear statement that IACUC review was not needed for the horse procedures in this study. They provide this in response to reviewer, but not the manuscript itself.
In the discussion, the authors stated, “These fidelity findings suggest that training non-mental health professionals to provide targeted interventions with supervision and support can be beneficial to youth mental wellness”. Since the authors did not include fidelity as a variable of analysis in client outcomes, can this claim be made?
In author response it is stated, “The second study was designed to assess the similar outcomes for the student riders, as well as extended to collect data from volunteers and horses.” What were the volunteers assessed on?
Author Response
Please see attached for a summary of edits and changes.
Thank you
